# Team DMG at CMCL 2022 Shared Task: Transformer Adapters for the Multi- and Cross-Lingual Prediction of Human Reading Behavior

**Ece Takmaz**

Institute for Logic, Language and Computation
University of Amsterdam
`ece.takmaz@uva.nl`

## Abstract

In this paper, we present the details of our approaches that attained the second place in the shared task of the ACL 2022 Cognitive Modeling and Computational Linguistics Workshop. The shared task is focused on multi- and cross-lingual prediction of eye movement features in human reading behavior, which could provide valuable information regarding language processing. To this end, we train 'adapters' inserted into the layers of frozen transformer-based pretrained language models. We find that multilingual models equipped with adapters perform well in predicting eye-tracking features. Our results suggest that utilizing language- and task-specific adapters is beneficial and translating test sets into similar languages that exist in the training set could help with zero-shot transferability in the prediction of human reading behavior.

## 1 Introduction

Eye movements provide valuable information about the contents of underlying cognitive processes and where our attention falls (Rayner, 1977). Predicting human reading behavior as reflected in eye movements is an important task that requires capturing universal aspects of language processing as well as its language-specific properties (Liversedge et al., 2016; Hollenstein et al., 2021b). This task could help us gain insight into language-related eye movements and the predictive capabilities of the models of human reading behavior.

Various approaches have been proposed for the modeling of human reading behavior (Rayner, 1998; Reichle et al., 1998; Hahn and Keller, 2016). The CMCL 2021 shared task focused on the prediction of 'monolingual' reading behavior and the participants applied various methodologies to predict eye-tracking features, e.g. gradient boosting, ensembling, using handcrafted features, deep learning (Hollenstein et al., 2021a; Bestgen, 2021; Li and Rudzicz, 2021; Oh, 2021; Vickers et al., 2021).

With regard to deep learning-based approaches, there exist findings suggesting that, as compared to transformer-based models (Vaswani et al., 2017), recurrent neural networks exhibit attention patterns closer to human attention (Sood et al., 2020). However, more recently, transformer-based models have been shown to better account for human reading behavior than recurrent neural networks (Merkx and Frank, 2021). Moreover, pretrained language models (PLM) such as BERT (Devlin et al., 2019) and XLM (Conneau and Lample, 2019) can predict multilingual human reading behavior well (Hollenstein et al., 2021b), in addition to having advanced the state-of-the-art in many downstream NLP tasks.

The focus of the CMCL 2022 shared task (Hollenstein et al., 2022) is to predict four eye-tracking features for data containing sentences in 6 different languages as well as transferring to a new language. For this purpose, we train 'adapters' inserted into transformer layers of frozen PLMs (Houlsby et al., 2019). We find that training adapters for each language separately within multilingual transformers leads to good performance, attaining the second place in the leaderboard. In addition, we show that such models can transfer to new languages via simply translating the new test sets into closely-related languages (e.g. lexically or grammatically) that the model was exposed to during training.[1]

## 2 Background

### 2.1 Data and Subtasks

The CMCL 2022 shared task consists of 2 subtasks. The data for Subtask 1 includes publicly-available eye-tracking corpora for 6 languages (English, Chinese, Russian, Hindi, German, Dutch). These corpora differ in size as well as the nature of the sentences they contain (i.e. news articles, scientific texts, Wikipedia entries). The data is already par-

---

[1] Our repository: `https://github.com/ecekt/cmcl2022_dmg`

titioned into train, validation and test splits. For Subtask 2, we are only supplied with a test set comprised of Danish sentences. We only use the data provided in the shared task and preprocess the textual input utilizing the tokenizers of PLMs. For more details, see Appendix A.

The eye-tracking features provided in the data correspond to 'first fixation duration' (FFD, duration of the first fixation on the current word) and 'total reading time' (TRT, total duration of all fixations on the current word including regressions). The values of these features were provided per token entry, averaged across all the readers: **FFDAvg** and **TRTAvg**. In addition, to account for the individual differences between readers, the data also includes the standard deviations of these features across readers: **FFDStd** and **TRTStd**.

The aim of the subtasks is to predict these 4 features for each token. The submissions are ranked with respect to test-set Mean Absolute Error (MAE): the average of the absolute differences between the ground-truth values and the values output by the model (see Appendix B). The shared task system also reports coefficients of determination ($R^2$), which we provide in Appendix F.

## 2.2 Adapters

The common method for using PLMs in downstream tasks is to fine-tune them for each task. If there are multiple tasks the model should handle at the same time, this could lead to some issues (Pfeiffer et al., 2021). For instance, learning tasks in parallel could cause interference and the model might learn a certain task better than the others. In the case of sequential training, we might observe catastrophic forgetting, where the model forgets the previously learned tasks. In addition, usually the whole model is fine-tuned; hence, we might need to save a new model per task, which increases compute and memory requirements.

To overcome these issues, 'adapters' have been proposed (Houlsby et al., 2019; Bapna and Firat, 2019). Adapters are bottleneck layers consisting of new weights integrated into each layer of a transformer model. They first project down ($W_D \in \mathbb{R}^{h \times d}$) the dimensions of the transformer hidden state $h_l$ at layer $l$, apply a non-linearity, and then project the activations back up ($W_U \in \mathbb{R}^{d \times h}$) to the original dimensions. The outcome is then summed up with the residual $r_l$ via a skip-connection to

obtain the output of the adapter $A_l$:

$$A_l = W_U(ReLU(W_D h_l)) + r_l \qquad (1)$$

Keeping the pretrained model frozen and only training adapters have been shown to yield performances close to those of fully-fine-tuned models while also maintaining efficiency (Houlsby et al., 2019; Bapna and Firat, 2019; Rücklé et al., 2021). Various types of adapters, insertion and training schemes have been proposed for machine translation, multi-task settings and cross-lingual transfer (Ansell et al., 2021; Pfeiffer et al., 2020b, 2021; Philip et al., 2020; Üstün et al., 2020, 2021; Poth et al., 2021).

Given their relevant advantages, we use Adapters from AdapterHub framework (Pfeiffer et al., 2020a)[2] built on HuggingFace Transformers (Wolf et al., 2020), to insert trainable adapters into frozen PLMs for the prediction of eye-tracking features. Then, we train language- and task-specific adapters and store their trained weights along with a single model. The details of the models and adapters used in Subtasks 1 and 2 are provided in Sections 3 and 4, respectively. For reproducibility, the hyperparameters for the best models selected with respect to their MAE scores on the validation set and the details of the development environment are provided in Appendices C and D.

## 3 Subtask 1: Multi-lingual

In this subtask, the aim is to predict eye-tracking features for data from 6 languages, for which we have training, validation and test sets. We focus on comparing a single setup for all languages vs. separate setups for different languages.

## 3.1 Methodology

**Single adapter for all languages**   We first train a single task-specific adapter integrated into a frozen PLM on all the languages per eye-tracking feature. We utilize the XLM-RoBERTa-base (XLM-R) model (Conneau et al., 2020), which is a multilingual version of RoBERTa (Liu et al., 2019), trained with the masked language modeling objective on 100 languages covering all of the shared task languages.[3]

We place a token-level regression head on top of XLM-R. We then train this head and the adapters

---

[2] https://adapterhub.ml
[3] https://huggingface.co/xlm-roberta-base

to predict eye-tracking features for each contextualized token in a given sentence. Since we keep the underlying model frozen, this method only learns a small set of parameters for the eye-tracking features, which we expect would capture universal patterns in human reading behavior.

**Language-specific adapters** When a single model is trained on multiple languages, its capacity for certain languages might decrease, which is called 'the curse of multilinguality' (Conneau et al., 2020; Pfeiffer et al., 2020b). To avoid this issue, we increase the language-specific capacity by training adapters separately for each language.

In this approach, we train a single adapter that is specific to a language-task pair (yielding $6*4 = 24$ adapters) integrated into frozen XLM-R. In addition, we also implement another setup where we *stack* language- and task-specific adapters on top each other (Pfeiffer et al., 2020b). In the latter setup, per language, we utilize a frozen language-specific adapter that was trained on Wikipedia articles with the masked language modeling objective, as provided on AdapterHub (Pfeiffer et al., 2020b, 2021).[4] We train the new task-specific adapter and the token regression head to predict eye-tracking features specific to each language. For Dutch, AdapterHub did not have a language adapter trained on Wikipedia; therefore, we only use a single new adapter.[5]

PLM tokenizers produce multiple wordpieces for some tokens. For such tokens, the models output predictions for each wordpiece. We calculate their average value and assign it as the prediction for the whole token entry. To explore whether the way the wordpieces are treated has an effect on accuracy, we also train and test the stacked setup only keeping the first wordpiece to represent the full token entry.

## 3.2 Results

In the top half of Table 1, we present the results for Subtask 1. Overall, our models outperform the mean baseline and seem to predict FFD features better than TRT features. XLM-R with new adapters trained from scratch on all languages

together performs the worst. XLM-R with new language-specific adapters further improves the results, in particular decreasing the MAE of features corresponding to averages.

The XLM-R setup that stacks adapters per language yields our best results for Subtask 1 achieving second place in the leaderboard of the shared task (MAE = 3.6533, our second submission). The breakdown of results per language is provided in Table 2 in Appendix E. It can be observed from this table that the model performs well for languages such as German and Dutch, yet struggles with languages such as Chinese and Russian, which could be due to the differences in their typologies, the nature of the corpora, vocabulary size and the issues that might have been caused by the multilinguality of the underlying PLM.

Finally, utilizing only the first wordpieces seems to degrade the performance across the features (MAE = 3.7261, our third submission). This finding indicates that retaining all wordpieces provides a better picture of the value to be predicted, as each wordpiece might contribute to the processing of the full token, affecting fixation duration times.

## 4 Subtask 2: Cross-lingual

For this subtask, we conduct various experiments to obtain results for the Danish test set in the absence of training and validation data in this language.

## 4.1 Methodology

**Zero-shot** We first feed the Danish test set directly into the XLM-R all-languages model. Since the adapters in this case are expected to have learned universal eye movement features and XLM-R includes Danish in its training, we expect to see this model to transfer well to Danish without being exposed to eye-tracking data in this language.

**Translate train** In this approach, we translate the training and validation set from their source language into the target language to be used in the training of a new model (Conneau et al., 2018). We have chosen English as the source language, as it constitutes almost half of the whole shared task data and XLM-R performs well in English (Conneau et al., 2020). We translate the English training and validation data word-by-word[6] into Danish

---

[4] https://adapterhub.ml/explore/text_lang/ The names of the language-specific adapters are '{x}/wiki@ukp', where {x} is to be replaced by the abbreviation corresponding to the language, e.g. 'en/wiki@ukp'.

[5] We also experiment with training two new adapters stacked together for Dutch to make the setups more comparable. See Appendix E for the outcomes of additional models including the use of RoBERTa and XLM-RoBERTa-large.

[6] Sentence-by-sentence translation could yield more reliable outcomes; however, it may cause issues in word order and count: source and translated text would need to be aligned.

| Model setup | FFDAvg | FFDStd | TRTAvg | TRTStd | MAE |
|---|---|---|---|---|---|
| All languages together | 3.1449 | 1.9697 | 6.4339 | 4.6253 | 4.0434 |
| Language-specific | 2.8563 | 1.9741 | 5.5682 | 4.6956 | 3.7736 |
| Language-specific-stack | 2.6086 | 1.9219 | 5.6542 | 4.4284 | **3.6533** |
| First wordpiece-only | 2.6876 | 1.9609 | 5.7059 | 4.5501 | 3.7261 |
| Zero-shot | 3.4955 | 2.7370 | 7.1336 | 7.1502 | 5.1291 |
| Translate train | 14.6278 | 4.4001 | 19.8624 | 14.2824 | 13.2932 |
| Translate test - EN | 13.7903 | 5.1338 | 20.9214 | 13.5084 | 13.3385 |
| Translate test - EN (without Provo) | 4.5843 | 3.9382 | 9.3022 | 6.8426 | 6.1668 |
| Translate test - DE | 5.4512 | 1.7349 | 6.9036 | 5.7730 | **4.9657** |
| Mean baseline | 5.6858 | 2.5395 | 8.8200 | 5.8877 | 5.7332 |

Table 1: Test set results for Subtask 1 and Subtask 2. The best models per subtask are indicated in bold.

using the MarianMT en-da model.[7] Since Adapter-Hub currently does not host a language-specific adapter for Danish, we do not implement stacking and only train task-specific adapters for Danish.

**Translate test**  In this setup, we translate the test set into a language for which we have training and validation data (Conneau et al., 2018) using MarianMT models. We first translate the Danish test set into English word-by-word. Using the best English model we obtained in Subtask 1, we generate predictions for the translated test set. In addition, we notice that the Provo corpus (Luke and Christianson, 2018) in the English subset has rather higher values for the features as compared to the other English corpora existing in the data. As a result, we retrain the best English setup using the same hyperparameters and skipping the Provo data.

In our final setup for Subtask 2, we translate Danish into German and utilize the best German model from Subtask 1 to obtain predictions. The main reason for opting for German was to better account for the effects of word order, e.g. inversions in main and subordinate clauses, exploiting the syntactic similarities between Danish and German.

### 4.2 Results

The bottom half of Table 1 provides the results for Subtask 2. First of all, the translate train approach does not seem to be a viable option, as its accuracy is much lower than the mean baseline (MAE = 13.2932, our first submission). Using the translate test approach in English yields very similar results. However, as we hypothesized, remov-

ing the Provo corpus from the training improves the translate test performance substantially (MAE = 6.1668, our second submission), albeit still underperforming. The zero-shot setup, on the other hand, yields a MAE score better than the mean baseline, suggesting that our adapters learn universal eye-tracking feature across languages combined with the multilingual pretraining of XLM-R.

Finally, the translate test setup in German yields our best results for this subtask achieving second place in the leaderboard (MAE = 4.9657, our third submission). These results indicate that the selection of source language and data has an effect on the results. Furthermore, it can be claimed that translate test is a viable option for adapters integrated into PLMs for achieving good transfer to a test set in a new language, without being exposed to actual eye-tracking data in this language.

## 5 Conclusion

We have trained language- and task-specific adapters for the prediction of eye-tracking features reflecting human reading behavior in multi- and cross-lingual settings. Our best models performed well, attaining the second place in the CMCL 2022 leaderboard. This suggests that pretrained language models enhanced with small adapter layers possess the capability to predict eye-tracking features.

In addition to our setups, other methods such as dropping adapters or adapter fusion could be implemented (Rücklé et al., 2021; Pfeiffer et al., 2021). It would also be informative to consider autoregressive models and the possibility of making use of various lexical and syntactic features and additional cognitive signals. The prediction of

---

[7]https://huggingface.co/docs/transformers/model_doc/marian

each eye-tracking feature could also be informed by other eye-tracking features, as each of them represents different aspects of human reading behavior. Similar approaches could also be of help in the modeling of other human cognitive signals, opening up novel ways of predicting and inspecting cognitive processes in humans.

## Acknowledgments

We would like to thank Raquel Fernández, Sandro Pezzelle and Ahmet Üstün for their valuable feedback regarding the project and the paper. This project has received funding from the European Research Council (ERC) under the European Union's Horizon 2020 research and innovation program (grant agreement No. 819455).

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

# Appendix

## A Data preprocessing

We use the XLM-RoBERTa tokenizer containing 250002 tokens. When converting the words into IDs, the tokenizer maintains the cases of the words, which could provide crucial information regarding human reading behavior. However, the way the tokens were presented to the readers differ from how the tokenizer would partition a given sentence. For instance, in the data, we see full stop appended to the last word or '(1917-1919)' as a single entry. For such cases, the tokenizer yields multiple wordpieces per token. We assign the eye-tracking feature values of the full entry to each of its wordpieces and during training and validation, we include them in the loss separately. For the test set predictions, we calculate the average of the predictions for the wordpieces and assign it as a single prediction for the whole entry.

We combine the token entries having the same sentence ID into a single sentence. Since the sentences do not include start- and end-of-sentence tokens, we also add such special tokens where necessary. In addition, we pad or truncate the input to maintain a total wordpiece length of 200. For all special tokens, we assign '-1' as the dummy eye-tracking feature value.

## B Metric

We implement MAE as below:

$$\frac{\sum_{i=1}^{N} |o_i - t_i|}{N} \tag{2}$$

where $N$ is the number of tokens in the data, $o_i$ is the value output by the model for a given token, and $t_i$ is the ground-truth value for this token. We calculate MAE for all 4 eye-tracking features and take their average to obtain the final MAE.

## C Hyperparameters

For each model, we have performed hyperparameter search for learning rate (0.001, 0.0001, 0.00001, 0.00002) and batch size (4, 8, 16, 32). All the models were trained up to 50 epochs.[8] We saved the best model based on the validation MAE per epoch and ran random initializations of the best model with 4 different seeds. The adapters were optimized using the AdamW optimizer (Loshchilov and Hutter, 2019) with respect to MSELoss following a linear learning rate schedule. In Table 3, we provide the hyperparameters of our best models for Subtask 1 and Subtask 2.

## D Environment details

We use AdapterHub version 2.2.0 based on HuggingFace Transformers version 4.11.3.[9] We implement and train our models in Python version 3.7.11 and PyTorch version 1.10.1.[10] All models were run on a computer cluster running Debian Linux OS, with 4 NVIDIA GeForce GTX 1080 Ti GPUs with driver version 470.103.01 and CUDA version 11.4.

## E More results

**RoBERTa + NER** Our first submission to Subtask 1 was built on RoBERTa-base (Liu et al., 2019),[11] with a Named Entity Recognition (NER) adapter trained on the CoNLL2003 dataset[12] (Poth et al., 2021; Tjong Kim Sang and De Meulder, 2003). We used the NER adapter as we noticed a lot of named entities in the data. In this setup, we remove the NER token classification head and create a token-level regression head. The head is trained from scratch and the NER adapter is fine-tuned. The results revealed that this setup already

---

[8]It is possible that a higher epoch cap could produce better results; however, in most cases, we observed declining performance as the number of epochs approached 50.

[9]https://huggingface.co/docs/transformers/

[10]https://pytorch.org/

[11]https://huggingface.co/docs/transformers/model_doc/roberta

[12]https://adapterhub.ml/adapters/AdapterHub/roberta-base-pf-conll2003/

| Model setup | FFDAvg | FFDStd | TRTAvg | TRTStd | MAE | Baseline MAE |
|---|---|---|---|---|---|---|
| EN stack | 3.2360 | 1.9582 | 6.8383 | 4.9501 | 4.2456 | 5.2736 |
| EN large stack | 3.0390 | 1.9921 | 6.1242 | 4.8968 | 4.0130 | |
| ZH stack | 3.1586 | 3.3608 | 6.8213 | 6.6955 | 5.0091 | 5.4616 |
| ZH large stack | 3.1571 | 3.4448 | 7.3876 | 6.5892 | 5.1447 | |
| DE stack | 0.4304 | 0.4346 | 3.7796 | 2.8918 | 1.8841 | 2.8679 |
| HI stack | 2.5493 | 2.7178 | 5.7471 | 5.5693 | 4.1459 | 4.5668 |
| RU stack | 2.6062 | 2.6443 | 8.3637 | 5.5609 | 4.7938 | 4.9007 |
| NL 1 new | 1.8772 | 1.5720 | 3.3467 | 2.9443 | 2.4351 | 2.4176 |
| NL 2 new stack | 1.8904 | 1.5911 | 3.2836 | 3.0673 | 2.4581 | |

Table 2: Test set results for Subtask 1 for the XLM-R language-specific models with stacking, broken down into languages. Baseline MAE is calculated with respect to the means of the language-specific data. EN: English, ZH: Chinese, DE: German, HI: Hindi, RU: Russian, NL: Dutch.

| Model | LR | Batch size | Seed |
|---|---|---|---|
| EN stack | 0.0001 | 4 | 42 |
| ZH stack | 0.001 | 4 | 8 |
| DE stack | 0.001 | 8 | 42 |
| HI stack | 0.001 | 4 | 42 |
| RU stack | 0.001 | 4 | 8 |
| NL 1 new | 0.001 | 4 | 42 |

Table 3: Hyperparameters for our best submission for Subtask 1 (Language-specific-stack). DE stack model is also used in obtaining our best results for Subtask 2. LR: Learning rate.

improves over the mean baseline across all features (MAE = 4.0317, our first submission). Although RoBERTa is monolingual (English) and its vocabulary is much smaller than XLM-R's vocabulary (50265, also its tokenizer converts non-Latin scripts into unintelligible wordpieces), this model seemed to work quite well. However, we wanted to make sure that the wordpieces work properly and that the underlying frozen PLM was exposed to multilingual data, which is why we switched to XLM-RoBERTa.

**Language breakdown** The details of the language-specific-stack models for Subtask 1 are provided in Table 2. The majority of these models outperform the corresponding mean baselines computed with respect to the language-specific means (except for the Dutch setup, which does not include a pretrained language-specific adapter).

**Dutch-specific models** For Dutch, we only employed a single adapter as we did not have a Dutch-specific adapter pretrained on Wikipedia articles. As a result, we also tried stacking 2 new adapters. This setup yielded slightly worse scores than the former setup. Therefore, we opted for keeping the single-adapter model in our submissions.

**Large models** We also use the large version of XLM-RoBERTa.[13] At the time of writing, only English and Chinese Wikipedia MLM adapters were available on AdapterHub (Pfeiffer et al., 2020b, 2021).[14] For English, the utility of the large model was not substantially high, and for Chinese, the large model caused a decrease in accuracy. These findings suggest that the adapters are able to capture the patterns in eye-tracking features, without the need to resort to larger language models. However, more hyperparameter tuning could be beneficial to explore the capacity of the large models.

## F $R^2$ scores

In Table 4, we provide the $R^2$ (coefficient of determination) scores as reported by the shared task system. The top half lists the results for Subtask 1 and the bottom half for Subtask 2.

---

[13]https://huggingface.co/xlm-roberta-large
[14]EN: https://adapterhub.ml/adapters/ukp/xlm-roberta-large-en-wiki_pfeiffer/, ZH: https://adapterhub.ml/adapters/ukp/xlm-roberta-large-zh-wiki_pfeiffer/

| Model | FFDAvg | FFDStd | TRTAvg | TRTStd | $R^2$ |
|---|---|---|---|---|---|
| RoBERTa + NER | 0.6963 | 0.3437 | 0.3293 | 0.2677 | 0.4093 |
| Language-specific-stack | 0.7581 | 0.3689 | 0.4868 | 0.3517 | 0.4914 |
| First wordpiece-only | 0.7506 | 0.3564 | 0.4836 | 0.3362 | 0.4817 |
| Translate train | -13.5708 | -3.1490 | -6.1914 | -5.4032 | -7.0786 |
| Translate test - EN (without Provo) | -1.0249 | -2.3468 | -0.8361 | -0.7824 | -1.2475 |
| Translate test - DE | -1.2176 | -0.1296 | -0.4203 | -0.4929 | -0.5651 |

Table 4: $R^2$ scores for the submissions to Subtask 1 and 2.