# OpenReview forum: "Team DMG at CMCL 2022 Shared Task: Transformer Adapters for the Multi- and Cross-Lingual Prediction of Human Reading Behavior"
_aclweb.org/ACL/2022/Workshop/CMCL_Shared_Task — CMCL Shared Task_

### Official Review · Reviewer_6qvh · 2022-03-25
**Team DMG at CMCL 2022 Shared Task**

**Rating:** 7
**Confidence:** 5

**Review:**

# Summary
In this work, the authors propose the application of adapters to predict first fixation duration (FFD) and total fixation duration (TFD) averaged across tokens.
The authors utilize frozen pre-trained language models in combination with bottleneck adapters to predict the properties mentioned before.
They investigate several approaches: e. g. fine-tuning, mono- and multilingual PLMs, training or test set translations.
The final approach placed second in the Shared Task.


# Pros
- A novel approach using frozen PLMs with adapters to predict token reading measures.
- Discuss various approaches with possible theoretical explanations.
- Reproducibility via publicly available code.
- Second place on the leaderboard.


# Cons
- None.


## Minor comments and missing references
Some minor suggestions/comments:

- Research by Sood et al. [1] suggests that  RNNs/CNNs resemble more human attention than transformer-based architectures.

- During the hyperparameter search, the epochs were always the maximum in the grid. My question is whether training further might have even further improved the performance?


### References
[1] Sood, Ekta and Tannert, Simon and Frassinelli, Diego and Bulling, Andreas. Interpreting Attention Models with Human Visual Attention in Machine Reading Comprehension. 2020. Proc. ACL SIGNLL Conference on Computational Natural Language Learning (CoNLL). doi:10.18653/v1/P17.

---

### Official Review · Reviewer_xo68 · 2022-03-27
**Review for Team DMG at CMCL 2022 Shared Task**

**Rating:** 7
**Confidence:** 5

**Review:**

# Summary
For their contribution, the authors used a pre-trained XLM-R language model and trained an adapter, inserted into the frozen pre-trained LM in order to predict eye-tracking reading measures (first fixation duration & total reading time). They test several methods for both subtask 1 (1 adapter for all languages, language-specific adapters) and subtask 2 (zero-shot, e.g. using the adapters from subtask 1, translation of training set, translation of test set).

# Pros
* Novel and efficient approach
* Good results (placed 2nd)
* The paper is nicely written and easy to follow
* Besides the task itself, the authors also address and comment on interesting conceptual issues (e.g. what types of patterns does a single vs. language-specific adapters capture)
* Translating the train/test sets is a simple (in a good way) yet effective approach for subtask 2

# Cons
* Since there seemed to be some space left, the methodology section could have been a bit more detailed or it could have been used for a formal definition of the problem setting & adapters.

---

### Decision · Program_Chairs · 2022-03-28

Accept